

# Effect of GR24 on the growth and development of licorice under low phosphorus stress

Yuting Jing[1,2], Man Li[1,2], Yong Wu[2], Chengming Zhang[2], Chengshu Qiu[2], Hengming Zhao[1,2], Li Zhuang[1] and Hongling Liu[2]

[1] Shihezi University, College of Life Sciences, Shihezi, Xinjiang, China
[2] Chengdu Normal University, Sichuan Provincial Key Laboratory for Development and Utilization of Characteristic Horticulural Biological Resources, Chengdu Normal University, Chengdu, Sichuan, China

## ABSTRACT

**Background.** *Glycyrrhiza*, a perennial herbaceous medicinal plant, is extensively utilized in the pharmaceutical industry. The growth of *Glycyrrhiza* is frequently constrained by soil phosphorus availability, as a significant portion of arable land in China suffers from phosphorus deficiency.

**Method.** This study utilized Ural *Glycyrrhiza uralensis* Fisch as the subject and examined the application of GR24, a synthetic strigolactone, under three phosphorus conditions: none (P1), low (P2), and high (P3). The research aimed to ascertain the optimal concentration of GR24 for promoting licorice growth and development, thereby providing a theoretical foundation for its agricultural management.

**Results.** The optimal GR24 concentration for P3 and P2 conditions was identified as G3 (10 μM), which enhanced growth metrics, chlorophyll a and b levels, while also boosting antioxidant enzyme activities in licorice. Specifically under P3, significant increases in liquiritigenin and glycyrrhizic acid levels were observed. Under P2, increases were noted in isoliquiritigenin, liquiritigenin, and liquiritin levels. Transcriptome analysis revealed differential gene, with 137 and 270 genes up-regulated and 77 and 294 genes down-regulated in the P3 and P2 treatments, respectively. GO functional enrichment identified 132 and 436 differentially expressed genes for P3 and P2 respectively, while KEGG pathways were predominantly enriched in plant-pathogen interactions and phenylpropanoid biosynthesis. Application of GR24 in P1 conditions did not significantly affect growth indices but did enhance glycyrrhetic acid, isoliquiritin, and liquiritin accumulation. Transcriptome profiling in this treatment identified 465 up-regulated and 1,109 down-regulated genes. GO annotation involved 1,108 differentially expressed genes, and KEGG analysis was primarily enriched in the plant-pathogen interaction pathway. Furthermore, transcription factor analysis revealed alterations in the C2H2, NAC, and MYB families, which are associated with phosphorus response.

Corresponding authors
Li Zhuang, 2214403407@qq.com
Hongling Liu, llhhll7878@163.com

## INTRODUCTION

Phosphorus is an essential nutrient that plays a pivotal role in nearly all metabolic processes within plants (*Kayoumu et al., 2023*). However, the concentration of effective phosphorus in the soil often fails to meet the demands of normal plant growth (*Qiu, Fan & Zeng, 2020*). Insufficient phosphorus supply can lead to significant changes in both the external and root morphology of plants, as well as transformations in their physiological characteristics (*Li et al., 2006*). As a non-renewable resource, global phosphorus reserves are limited, making soil phosphorus availability the primary limiting factor for high agricultural yields in China (*Tian, 2001*). The phosphorus required for plant growth and development is primarily sourced from soil phosphorus reserves and fertilization, ensuring adequate absorption by plants. The practice of increasing phosphorus fertilizer use, however, represents a "high input, low output" strategy. Consequently, maintaining high crop yields while protecting the environment has become a critical area of global research (*Yuan, He & Su, 2024*). Additionally, phosphorus stress significantly limits the production of medicinal plants, further emphasizing its crucial role in agricultural productivity (*Vance, Uhde-stone & Allan, 2022*).

*Glycyrrhiza uralensis* Fisch, a member of the genus *Glycyrrhiza Linn* within the *Leguminosae* family, is recognized for its medicinal properties, including pain relief, phlegm expulsion, cough suppression, energy enhancement, spleen tonification, and the modulation of various pharmaceuticals (*Gao et al., 2009*). Additionally, this species plays a crucial role as a sand-fixing plant in the desert and semi-desert regions of western China (*Du, 2007*). Nevertheless, over-harvesting has severely depleted wild licorice populations, rendering their resources critically scarce. Cultivated licorice, despite being a mainstream commodity, faces challenges such as inhibited growth, quality degradation, and reduced yields, issues that complicate adherence to the quality standards established in the Chinese Pharmacopoeia (*Gao, 2019*). Furthermore, the biochemical composition of licorice includes a diverse array of terpenoids (*Zhang & Deng, 2015*). Phosphorus plays a critical role in the biosynthesis of terpene precursors through the MVA pathway, involving acetyl-CoA, ATP, and NADPH, as well as through the MEP pathway, glyceraldehyde phosphate and pyruvate, both of which are significant for plant metabolic functions (*Kapoor et al., 2017*).

Strigolactones (SLs), sesquiterpene hormones derived from beta-carotene, are believed to play a crucial role in regulating both aboveground plant architecture and root development (*Omoarelojie et al., 2021*; *Ma et al., 2017*; *Marzec, 2016*). GR24, a synthetic Strigolactone, is recognized for its involvement in responses to abiotic stresses and acts as a positive regulator of stress responses (*Shi et al., 2024*). Specifically, GR24 has been shown to enhance drought and salt tolerance in *Arabidopsis thaliana* (*Yun et al., 2023*) and to interact with other hormones to promote lateral root growth in oilseed rape (*Ma et al., 2020*). Additionally, it has been demonstrated that exogenous GR24 can improve the metabolism of antioxidant enzyme systems, phenylpropanoids, nitric oxide (NO), and hydrogen sulfide (H2S) in strawberries, thereby maintaining fruit quality during storage (*Huang et al., 2021*). Moreover, dulcitolactone has been shown to regulate the plant type of medicinal plants,

thus influencing the growth and development of medicinal parts in a targeted manner to achieve the desired "optimal type" (*Cao et al., 2023*).

In the present study, our aim was to identify a more effective regimen for the growth of *Glycyrrhiza uralensis* Fisch in low-phosphorus environments and to investigate the effects of the exogenous application of Strigolactones on the accumulation of medicinal components.

# EXPERIMENTAL METHODOLOGY

## Overview of the pilot area

The experimental material consisted of *Glycyrrhiza uralensis* Fisch seedlings, each with four true leaves. These seedlings were cultivated and treated in the Wenjiang District of Chengdu located at coordinates 30°36′–30°52′N and 103°41′–103°55′E. This region is characterized by a subtropical monsoon climate, which offers a temperate environment, extended summer seasons, brief winter periods, and prolonged frost-free intervals (*Deng, 2022*).

## Experimental material

For the experiment, uniform and fully developed seeds of *Glycyrrhiza uralensis* Fisch were selected. These seeds were treated by soaking in 98% sulfuric acid for thirty minutes followed by extensive rinsing (more than three times) with distilled water (*Deng, 2022*; *Liang et al., 2016*). The substrate for the test was a homogeneous mixture of field soil and sand in a 1:1 ratio. The culture containers used were autoclaved hydroponic boxes (121 °C for 2 h) and 30 × 25 cm plastic pots, totaling 70 boxes.

## Drugs and reagents

The synthetic analog of Strigolactones, GR24, was procured from Beijing Kulaibo Science and Technology Co., Ltd., with a CSA No. of 76974-79-3. Initially, 1 mg of GR24 was dissolved in a small volume of acetone, and then diluted with water to a volume of 335.2 μL to prepare a 10 mM stock solution (*Zhu et al., 2022*). This stock solution was further diluted to create standard solutions at concentrations of 0 μmol/L, 1 μmol/L, 10 μmol/L, 100 μmol/L, and 1,000 μmol/L for experimental use.

## Experimental design

The experiment commenced on April 10, 2023, with the sterilization of licorice seeds followed by their germination in Petri dishes. Seven days post-germination, seedlings with a high top cover were transferred into hydroponic boxes (70 boxes, each containing nine plants, totaling 630 seedlings). These were cultivated hydroponically until the seedlings developed 4–5 true leaves, approximately 20 days later. Subsequently, seedlings exhibiting similar growth were selected and relocated to plastic pots measuring 30 cm in diameter and 25 cm in height. The cultivation substrate in these pots consisted of a sterilized mixture of garden soil and sand in a 1:1 ratio. Three seedlings were transplanted into each pot, resulting in a total of 120 pots. After transplantation, the pots were initially stored inside a building. One week later, they were moved to an outdoor experimental field protected from

rain. Soil culture began with the application of 1/4 strength Hoagland nutrient solution to gradually acclimate the seedlings over five days, followed by 1/2 strength for the next five days. Full-strength Hoagland solution was then used for regular management. When the seedlings bore 6–8 true leaves, they underwent a five-day starvation period during which they were moderately rehydrated (*Zhang et al., 2023*). Following the starvation treatment (*Zhang et al., 2023*), different phosphorus concentrations were administered. Three treatments were established: full-strength Hoagland solution as the high phosphorus supply (P3), one-third the original $NH_4H_2PO_4$ concentration in the Hoagland solution for the low phosphorus treatment (P2), and a phosphorus-free Hoagland solution for the zero phosphorus treatment (P1). Regular fertilization involved weekly watering with 500 ml of solution, supplemented by moderate additional watering as necessary. Foliar spraying with various concentrations of GR24 commenced on June 10. The treatment groups included not sprayed with GR24 (G1) and four additional groups treated with GR24 at concentrations of 1 μM (G2), 10 μM (G3), 100 μM (G4), and 1,000 μM (G5). Spraying was performed each evening until the leaves retained water droplets without dripping, repeated weekly. To minimize environmental biases, the placement of pots within each treatment group was randomized, and their positions were systematically rotated every two weeks. At the conclusion of the growing season in October 2023, the morphological and physiological indices of *Glycyrrhiza uralensis* were systematically measured.

## Indicators and methods
### Measurement of growth indicators
At the end of October 2023, all plants were harvested, and the adhering soil was gently rinsed off with running water. In each treatment group, five plants were sampled. The length of the underground portion was measured using a steel ruler, while the diameter of the main root was gauged with a vernier caliper. The fresh weights of the aboveground and underground parts of *Glycyrrhiza uralensis* Fisch were determined using a balance. A photograph of each specimen, with a ruler for scale, was taken and uploaded to Image J for root area projection analysis. The samples were then dried in an oven at 75 °C until a constant weight was achieved, and the dry weights of the aboveground and underground parts were subsequently weighed.

### Determination of chlorophyll content
For each treatment, leaves were randomly collected from five plants, washed, and dried. The veins were removed, and the remaining leaf tissue was chopped. A sample weighing 0.1 g was used for the extraction and quantification of chlorophyll a and b using the acetone method (*Yang, 1996*). The concentrations of chlorophyll a and b were calculated using the appropriate formulas.

### Antioxidant enzyme assay
From each treatment group, five plants were randomly selected. The fully expanded apical leaves were harvested, washed, dried, and weighed 0.1 g. The activities of superoxide dismutase (SOD) and peroxidase (POD) were measured spectrophotometrically using a Solarbio kit (Beijing Solarbio Science & Technology Co., Ltd., Beijing, China). Catalase

(CAT) activity was determined using a Shimadzu UV-2041 ultraviolet spectrophotometer (Shimadzu, Kyoto, Japan) (*Xu et al., 2022*).

### Materials and methods for the determination of pharmaceutical ingredients

The medicinal constituents of *Glycyrrhiza uralensis* Fisch were quantified using high-performance liquid chromatography (HPLC) as detailed by *Xu & Ma (2021)*. Standards for glycyrrhetic acid were sourced from Chengdu Pufide Biotechnology Co., isoliquiritin from Shanghai McLean Company, and other standards from Chengdu Kangbang Biotechnology Co. Ltd. Batch numbers for these standards include glycyrrhizic acid (21080201), glycyrrhetic acid (20041002), isoliquiritigenin (21101901), isoliquiritin (C11602211), glabridin (21032701), liquirtigenin (22110902), and liquirtin (21041301). Further details are provided in the Appendix.

### Transcriptome assays

The root tips of *Glycyrrhiza uralensis* Fisch were excavated, washed to remove soil, and immediately immersed in liquid nitrogen for rapid freezing. Subsequently, samples were stored at −80 °C and sent to Beijing Baimaike Company for transcriptome analysis.

## Data processing

Data were analyzed using SPSS 25 software. The significance of differences was assessed using Duncan's multiple range test and LSD test within a one-way ANOVA framework. Results were graphically represented using Origin2022 software.

## RESULTS AND ANALYSIS

### Effect of GR24 on growth indices of *Glycyrrhiza uralensis* Fisch under different phosphorus concentrations

Table 1 demonstrates that under no phosphorus stress, the fresh weight, dry weight, root length, basal stem diameter, and root projected area of *Glycyrrhiza uralensis* Fisch were lower compared to those observed under low phosphorus stress and high phosphorus supply treatments. Furthermore, these indices under low phosphorus stress were also reduced when compared to the high phosphorus supply conditions, that there was no difference between these indices. The variation in root projected area, however, was significant. There were no significant differences in any indices among the GR24 concentration treatments under no-phosphorus stress when compared to the G1 concentration treatments. Under low phosphorus stress, the G3 concentration notably enhanced the fresh weight, dry weight, root length, and projected root area of licorice by 53.8%, 38.2%, 20.1%, and 28.3%, respectively. In the high phosphorus supply treatment, the G3 concentration significantly increased the fresh weight, dry weight, root length, and basal stem diameter of *Glycyrrhiza uralensis* Fisch by 78.57%, 82.1%, 36%, and 45.8%, respectively. Moreover, the root projected area significantly increased by 46.1% and 36.1% under G3 and G5 concentrations, respectively.

**Table 1  Growth indicators of licorice in different treatment groups.**

|  |  | Fresh weight (g) | Dry weight (g) | Root length (cm) | Basal stem (mm) | Root projection area (cm²) |
|---|---|---|---|---|---|---|
| P1 | G1 | 1.73 ± 0.35cd | 0.65 ± 0.12ef | 20.32 ± 1.64ef | 3.04 ± 1.04cd | 6.65 ± 1.13 h |
|  | G2 | 1.68 ± 0.26cd | 0.63 ± 0.04f | 21.78 ± 2.20ef | 3.03 ± 0.65cd | 7.15 ± 0.75 h |
|  | G3 | 2.27 ± 0.50b | 0.80 ± 0.12def | 24.40 ± 3.48def | 3.36 ± 1.32bcd | 7.42 ± 0.97 h |
|  | G4 | 1.53 ± 0.21d | 0.67 ± 0.05ef | 18.76 ± 5.41f | 2.70 ± 0.48d | 6.33 ± 0.72 h |
|  | G5 | 1.92 ± 0.63bcd | 0.90 ± 0.11cdef | 22.44 ± 3.17ef | 2.72 ± 0.50bcd | 7.65 ± 1.17 h |
| P2 | G1 | 2.03 ± 0.43bcd | 0.66 ± 0.08ef | 24.88 ± 2.38cdef | 2.93 ± 0.39cd | 9.65 ± 0.29 g |
|  | G2 | 2.60 ± 0.74bc | 1.03 ± 0.12bcd | 32.50 ± 7.19bc | 3.29 ± 0.33bcd | 11.49 ± 0.85ef |
|  | G3 | 2.70 ± 0.88ab | 1.20 ± 0.27bc | 40.36 ± 9.73a | 3.71 ± 0.50bcd | 12.87 ± 0.98de |
|  | G4 | 1.75 ± 0.46bcd | 0.97 ± 0.20cdef | 31.7 ± 7.31bcd | 3.39 ± 0.45bcd | 9.95 ± 1.61fg |
|  | G5 | 2.41 ± 0.40bcd | 0.98 ± 0.21cde | 33.14 ± 7.82b | 2.94 ± 0.83cd | 11.02 ± 0.53fg |
| P3 | G1 | 1.87 ± 0.69bcd | 1.16 ± 0.38bc | 31.84 ± 4.30bcd | 3.15 ± 0.71cd | 11.46 ± 0.91ef |
|  | G2 | 2.57 ± 1.12bc | 1.36 ± 0.40b | 32.76 ± 5.95bc | 4.39 ± 1.11ab | 15.96 ± 1.32b |
|  | G3 | 3.48 ± 0.74a | 1.80 ± 0.40a | 43.68 ± 9.20a | 5.29 ± 0.88a | 17.65 ± 2.05a |
|  | G4 | 2.29 ± 0.59bcd | 1.05 ± 0.33bcd | 23.7 ± 3.48ef | 4.35 ± 1.29ab | 13.97 ± 1.13cd |
|  | G5 | 2.53 ± 0.74bc | 1.33 ± 0.22b | 27.34 ± 2.21bcde | 4.08 ± 0.74bc | 15.31 ± 1.30bc |

**Notes.**

Description: The data are average ± standard error (SD). P1: phosphorus-free treatment; P2: low phosphorus stress treatment; P3: high phosphorus supply treatment; G1: 0 μM GR24; G2: 1 μM GR24; G3: 10 μM GR24; G4: 100 μM GR24; G5: 1,000 μM GR24. Different lower-case letters after the same line mean significant difference in fresh weight, dry weight, root length, basal stem and root projection area ($P < 0.5$).

### Effect of GR24 on chlorophyll content of *Glycyrrhiza uralensis* Fisch at different phosphorus concentrations

As illustrated in Fig. 1, the chlorophyll content in the high phosphorus supply treatment group was higher than that observed in the low phosphorus stress and no phosphorus stress treatment groups, with the low phosphorus stress group exhibiting higher levels than the no phosphorus stress group. Notably, there was no consistent pattern in the changes of chlorophyll b content across the various phosphorus concentration treatments.

Under low phosphorus conditions, chlorophyll a content in the G3, G4, and G5 treatments was significantly higher than in the G1 concentration treatment, with increases of 50.07%, 28.53%, and 39.33% respectively. In the high phosphorus supply treatment, chlorophyll a content was significantly greater in the G2 and G3 concentration treatments compared to the G1 treatment, showing increases of 20.4% and 34.72% respectively. Regarding chlorophyll b content under low phosphorus stress, the G2 and G3 concentration GR24 treatments exhibited significantly higher levels than the G1 treatment, with increases of 7.06% and 18.17% respectively. Under high phosphorus supply conditions, chlorophyll b content in the G2, G3, and G4 concentration treatments was significantly greater than that in the G1 concentration treatment, with increases of 38.58%, 79.41%, and 69.46% respectively.

### Effect of GR24 on antioxidant enzyme activities of *Glycyrrhiza uralensis* Fisch at different phosphorus concentrations

Figure 2 illustrates the impact of phosphorus stress on the activities of superoxide dismutase (SOD), peroxidase (POD), and catalase (CAT). The activities of SOD, POD, and CAT

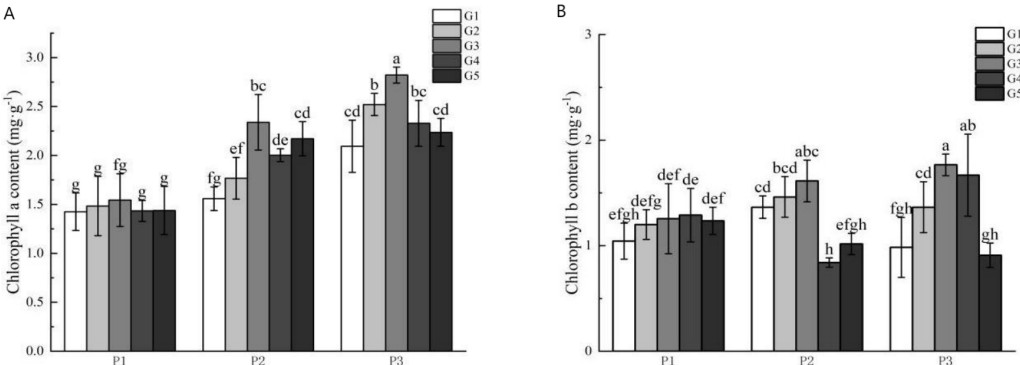

Fig.2 Chlorophyll content of licorice in different treatment groups

**Figure 1** **G Chlorophyll content of licorice in different treatment groups.** (A) Chlorophyll a; (B) Chlorophyll b. P1: phosphorus-free treatment; P2: low phosphorus stress treatment; P3: high phosphorus supply treatment; G1: 0 μM GR24; G2: 1 μM GR24; G3: 10 μM GR24; G4: 100 μM GR24; G5: 1,000 μM GR24. Different letters in the bars indicate significant differences according to Duncan's multiple range test and LSD test within a one-way ANOVA framework ($\alpha = 0.05$); $n = 5$ standard error.

increased with escalating phosphorus stress levels. The activity levels of SOD and POD under no phosphorus stress were higher than those in the low phosphorus stress group and the high phosphorus supply treatment group. Furthermore, CAT activities were significantly higher than those seen in the low phosphorus stress group and the high phosphorus supply treatment group. Also, SOD activities were higher in the no phosphorus stress group than in the highphosphorus supply treatment group, whereas POD and CAT activities were significantly higher in the low phosphorus stress group compared to the high phosphorus supply group.

Under no-phosphorus stress treatment, the G2 concentration of GR24 treatment significantly increased SOD activity by 16%, although the G3 concentration group showed no significant effect. There was also a significant increase in POD and CAT activities in the G2-G5 treatments compared to the G1 concentration treatment. Under low phosphorus stress treatment, G3 concentration treatment resulted in a significant 19.6% increase in POD activity compared to the G1 concentration treatment. Similarly, SOD activity in the G3 concentration treatment was significantly higher than in the G1 treatment, with an increase of 38.5%. The POD activity of the G3 concentration treatment also showed a significant increase of 22.9% over the G1 treatment. However, there were no significant improvements in CAT activity across any concentration treatment groups.

## Effect of GR24 on the content of medicinal components of *Glycyrrhiza uralensis* Fisch under different phosphorus concentrations

The experimental results indicated that the G3 concentration was most effective in regulating *Glycyrrhiza uralensis* Fisch; therefore, this concentration was selected for subsequent experiments. As summarized in Fig. 3 there were no significant differences in the levels of glycyrrhetic acid, liquirtigenin, liquirtin, and glabridin between treatments across varying phosphorus levels and GR24 concentrations. However, the content of

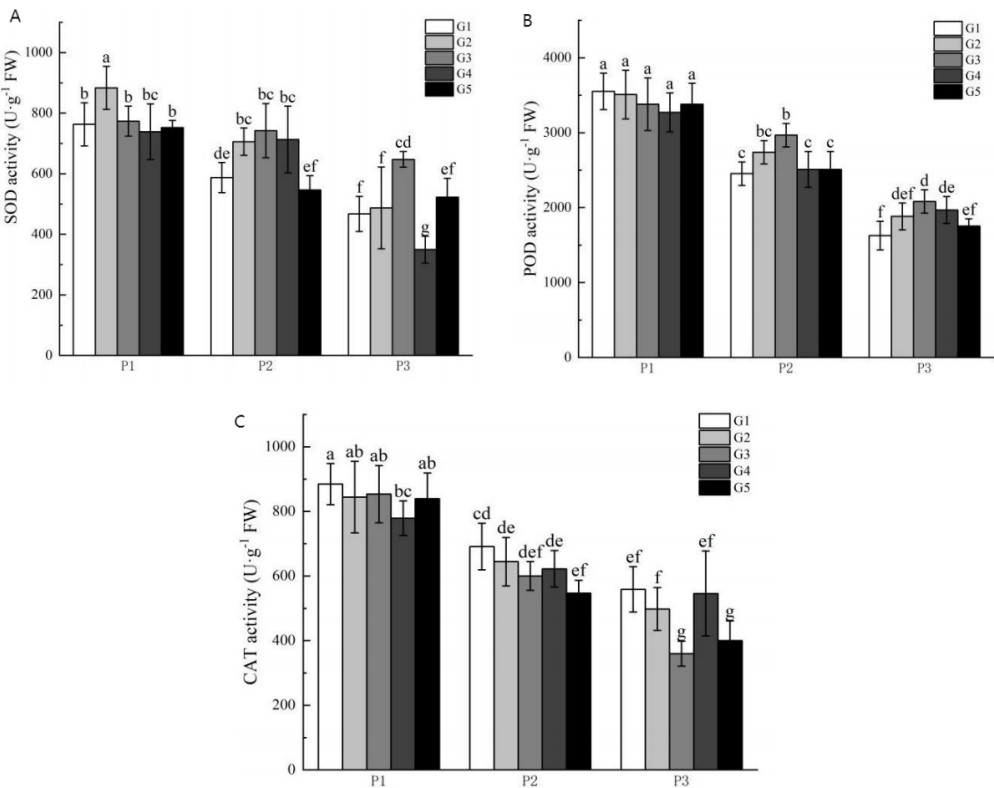

**Figure 2 Antioxidant enzyme activity of licorice in different treatment groups.** (A) SOD; (B) POD; (C) CAT. P1: phosphorus-free treatment; P2: low phosphorus stress treatment; P3: high phosphorus supply treatment; G1: 0 µM GR24; G2: 1 µM GR24; G3: 10 µM GR24; G4: 100 µM GR24; G5: 1,000 µM GR24. Different letters in the bars indicate significant differences according to Duncan's multiple range test and LSD test within a one-way ANOVA framework ($\alpha = 0.05$); $n = 5$ standard error.

isoliquirtigenin in the low phosphorus stress treatment was significantly higher than in the no-phosphorus stress treatment at both G1 and G3 concentrations. Similarly, isoliquirtigenin levels in the high phosphorus supply treatment were significantly elevated compared to those in the no-phosphorus stress treatment at both concentrations. Additionally, the levels of liquirtigenin in both the high phosphorus supply and low phosphorus stress treatments were lower than those in the no phosphorus stress treatment. Specifically, in the G3 concentration treatment, the contents of liquirtigenin and glycyrrhizic acid increased significantly by 72.2% and 21.3%, respectively. Under low phosphorus stress, the levels of isoliquirtigenin, liquirtigenin, and liquirtin were significantly higher in the G3 treatment than in the G1 treatment, showing increases of 131.3%, 118.8%, and 145.8%, respectively. In the phosphorus-free treatment, the G3 concentration significantly boosted the contents of isoliquirtigenin and liquirtin by 164.5% and 23.9%, respectively.

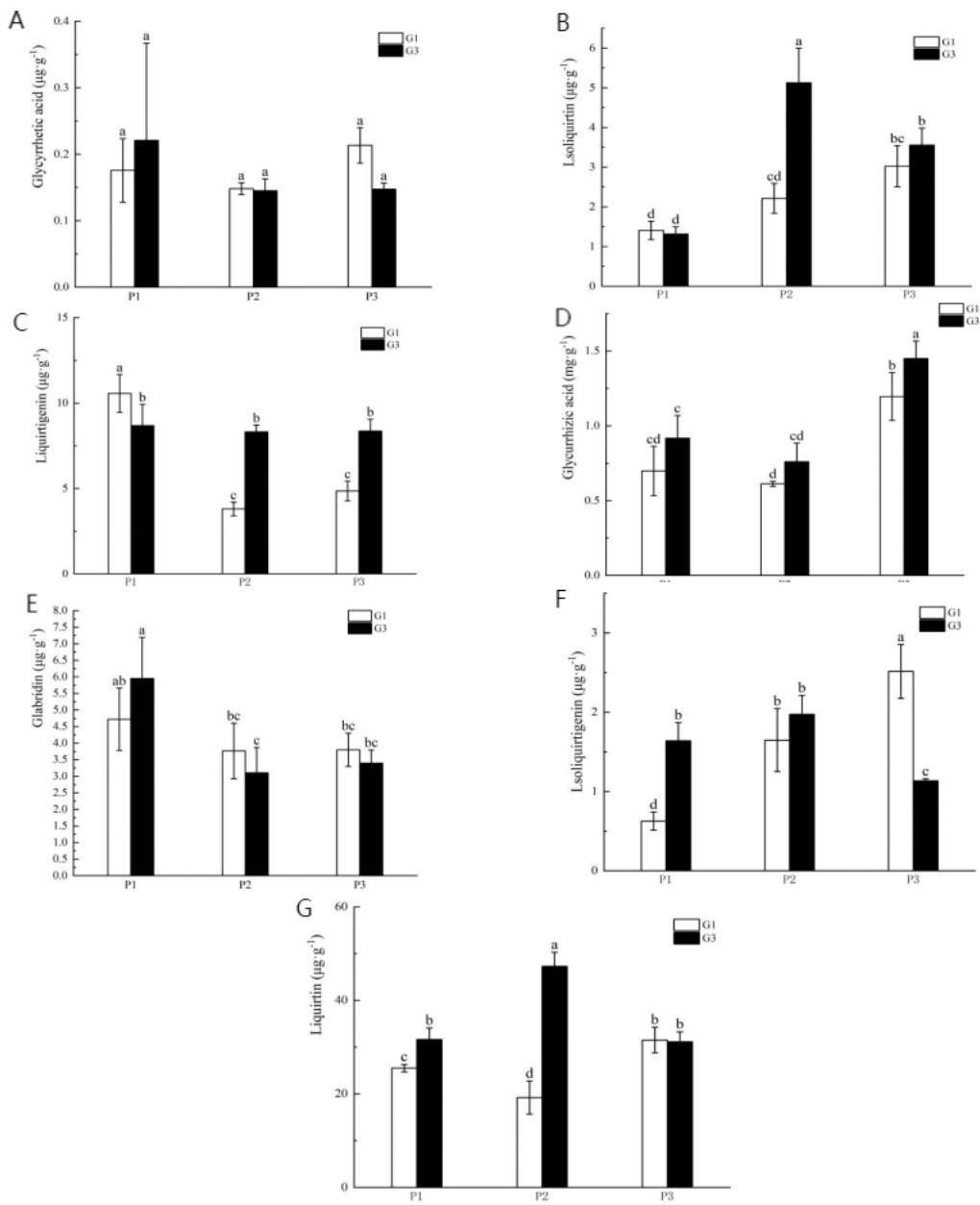

**Figure 3   Content of medicinal components in licorice from different treatment groups.** (A) Glycyrrhetic acid; (B) isoliquiritin; (C) Liquirtigenin; (D) glycurrhizic acid; (E) glabridin; (F) lsoliquirtigenin; (G) liquirtin. P1: phosphorus-free treatment; P2: low phosphorus stress treatment; P3: high phosphorus supply treatment; G1: 0 µM GR24; G2: 1 µM GR24; G3: 10 µM GR24; G4: 100 µM GR24; G5: 1,000 µM GR24. Different letters in the bars indicate significant differences according to Duncan's multiple range test and LSD test within a one-way ANOVA framework ($\alpha = 0.05$); $n = 5$ standard error.

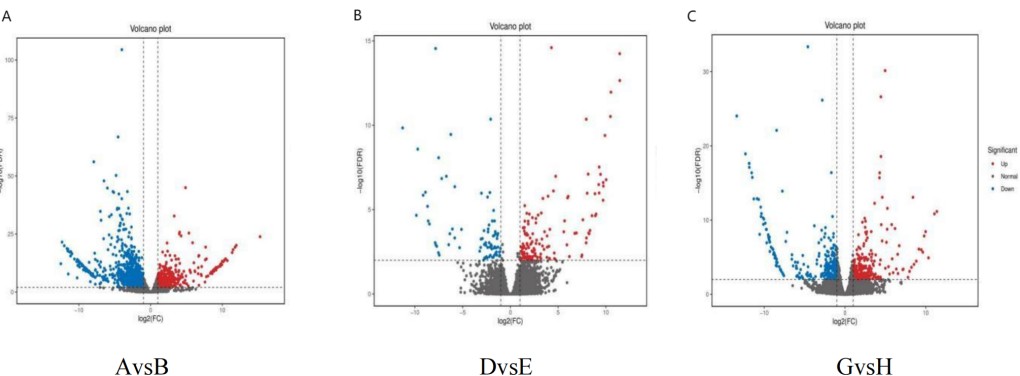

**Figure 4** **Differentially expressed genes.** (A) AvsB (P1G1 *vs* P1G3); (B) DvsE (P2G1 *vs* P2G3); (C) GvsH (P3G1 *vs* P3G3). P1: phosphorus-free treatment; P2: low phosphorus stress treatment; P3: high phosphorus supply treatment; G1: 0 μM GR24; G2: 1 μM GR24; G3: 10 μM GR24; G4: 100 μM GR24; G5: 1,000 μM GR24. Each point in the volcano plot of differential expression represents a gene, and the horizontal axis represents the logarithmic value of the expression level difference of a certain gene in two samples. The vertical axis represents the statistically significant negative logarithm of changes in gene expression levels. The larger the absolute value of the horizontal axis, the greater the difference in expression levels between the two samples; The larger the vertical axis value, the more significant the differential expression, and the more reliable the differentially expressed genes selected. The blue dots in the figure represent down regulated differentially expressed genes, the red dots represent up regulated differentially expressed genes, and the gray dots represent non differentially expressed genes.

## Effect of GR24 on the expression of *Glycyrrhiza uralensis* Fisch genes under different phosphorus concentrations
### Transcriptome sequencing quality assessment

In this study, RNA sequencing analysis was performed on 18 samples, yielding a total of 122.60 Gb of clean data, with each sample clean data approximately 5.82 Gb (see Table S3). The quality of the sequencing was substantiated by a Q30 base percentage exceeding 95.17% (see Table S4), reflecting reliable and accurate base recognition. The clean reads from each sample were mapped against the *Glycyrrhiza* designated reference genome with a matching efficiency ranging from 70.36% to 91.75%.

### Differential gene expression analysis

Differential gene expression analysis was conducted for the comparisons AvsB (G1 and G3 under no phosphate stress), DvsE (G1 and G3 under low phosphate stress), and GvsH (G1 and G3 under high phosphate) using a fold change threshold of $\geq 2$ and a FDR of <0.01 (Fig. 4). The statistical power for AvsB was 0.7422, DvsE was 0.4688, and GvsH was 0.506. A total of 1,574 differential gene expressions were identified in AvsB, comprising 465 up-regulated and 1,109 down-regulated genes. In DvsE, 214 differential expressions were identified, with 137 up-regulated and 77 down-regulated genes. For GvsH, 588 differential expressions were noted, evenly split with 294 up-regulated and 294 down-regulated genes.

### GO enrichment analysis

The GO functional enrichment analysis of DEGs using InterProScan in the *Glycyrrhiza uralensis* Fisch transcriptome, as depicted in Fig. 5, revealed that 1,108, 132, and 436 DEGs

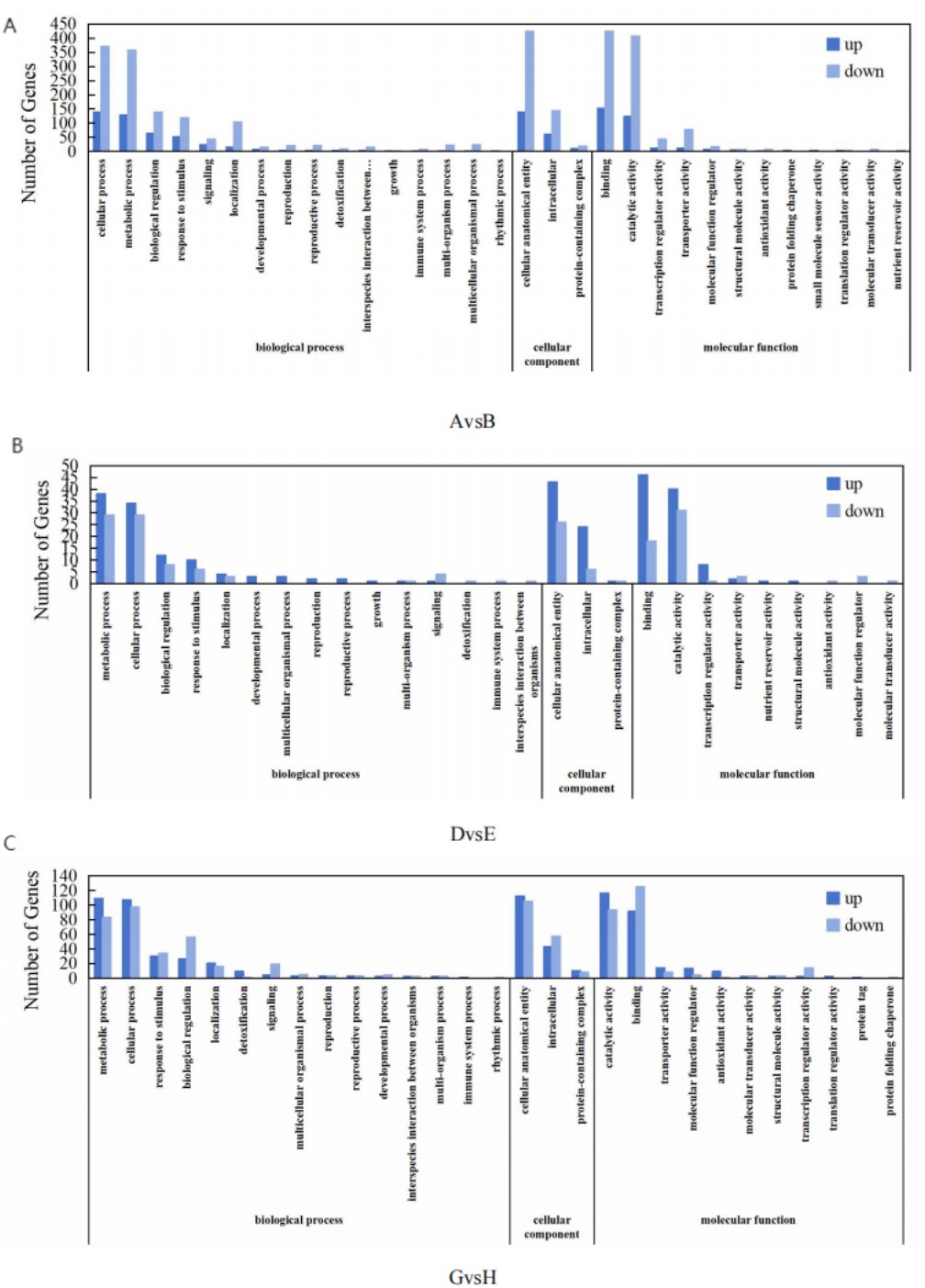

**Figure 5** **COG functional distribution map of differentially expressed genes.** (A) AvsB (P1G1 *vs* P1G3); (B) DvsE (P2G1 *vs* P2G3); (C) GvsH (P3G1 *vs* P3G3). P1: phosphorus-free treatment; P2: low phosphorus stress treatment; P3: high phosphorus supply treatment; G1: 0 µM GR24; G2: 1 µM GR24; G3: 10 µM GR24; G4: 100 µM GR24; G5: 1,000 µM GR24. The horizontal axis represents GO classification, and the vertical axis represents the number of genes. Blue represents the number of upregulated genes between the two groups, while light blue represents the number of downregulated genes between the two groups.

were annotated in the GO database for the comparative analyses of AvsB, DvsE, and GvsH, respectively. In the AvsB comparison, DEGs were classified into 17 categories of biological processes (BP), three of cellular components (CC), and 12 of molecular functions (MF). DvsE showed distributions across 16 BP classes, three CC classes, and 11 MF classes, whereas GvsH had 19 BP classes, three CC classes, and 11 MF classes, with an additional category in MF bringing the total to 13. Further scrutiny of the highest-ranked GO terms indicated a primary concentration in BP, especially under the AvsB treatment. Notably, the cellular process was significantly enriched with 509 alterations, comprising 139 up-regulations and 370 down-regulations. Similarly, the cellular anatomical entity exhibited 560 modifications, with 138 up-regulations and 423 down-regulations, aligning with the enrichment in CC, which also displayed 560 changes. In the DvsE set, the metabolic process in BP included 67 changes (38 up-regulated and 29 down-regulated), and the cellular anatomical entity in MF showed 69 changes with an equal number of up-regulations. Additionally, the catalytic activity in CC showed 71 changes, with 40 up-regulations and 31 down-regulations. For GvsH, the analysis reflected 204 changes in the cellular process of BP (107 up-regulated and 97 down-regulated), 217 changes in the cellular anatomical entity of MF (112 up-regulated and 105 down-regulated), and in CC, binding was characterized by 216 alterations (91 up-regulated and 125 down-regulated).

### KEGG pathway enrichment analysis

KEGG pathway enrichment analysis was conducted on DEGs to identify the most significant metabolic and regulatory pathways affected. As illustrated in Fig. 6, the analysis highlighted 20 pathways with the highest enrichment levels across the comparisons of AvsB, DvsE, and GvsH. In the AvsB and DvsE comparisons, the plant-pathogen interaction pathway was predominant, with 96 and 18 entries, respectively. This was followed by the GvsH comparison in the MAPK signaling pathway-plant with 36 entries and the Phenylpropanoid biosynthesis (PBP) pathway with 14 entries. Additionally, the DNA replication pathway was notable with 8 entries. AvsB showed a significant enrichment of four genes in the Flavone and flavonol biosynthesis pathway. Similarly, in GvsH, there was a notable enrichment of four genes in the Flavonoid biosynthesis pathway and two genes in the Terpenoid backbone biosynthesis pathway, with another five and four genes enriched in the latter pathway respectively.

### Analysis of differential transcription factors

The *Glycyrrhiza uralensis* Fisch glabra transcriptome data revealed a total of 1,495 unigenes annotated as transcription factors. As summarized in Table 2, 24 types of transcription factors exhibited varied expression patterns across the AvsB, DvsE, and GvsH comparisons. The C2H2, NAC, WRKY, MYB, and GRAS transcription factors showed the highest numbers of differentially expressed genes. Specifically, the most up-regulated transcription factor families in AvsB included AP2/ERF-ERF, C2H2, and MYB, with up-regulations noted for six, two, and two genes, respectively. The most down-regulated families were C2H2, NAC, and WRKY, with 10, nine, and eight genes down-regulated, respectively. In DvsE, the top up-regulated transcription factor families were WRKY, MYB, and bHLH, each with increases in two, one, and one genes, respectively; no down-regulations were observed. For

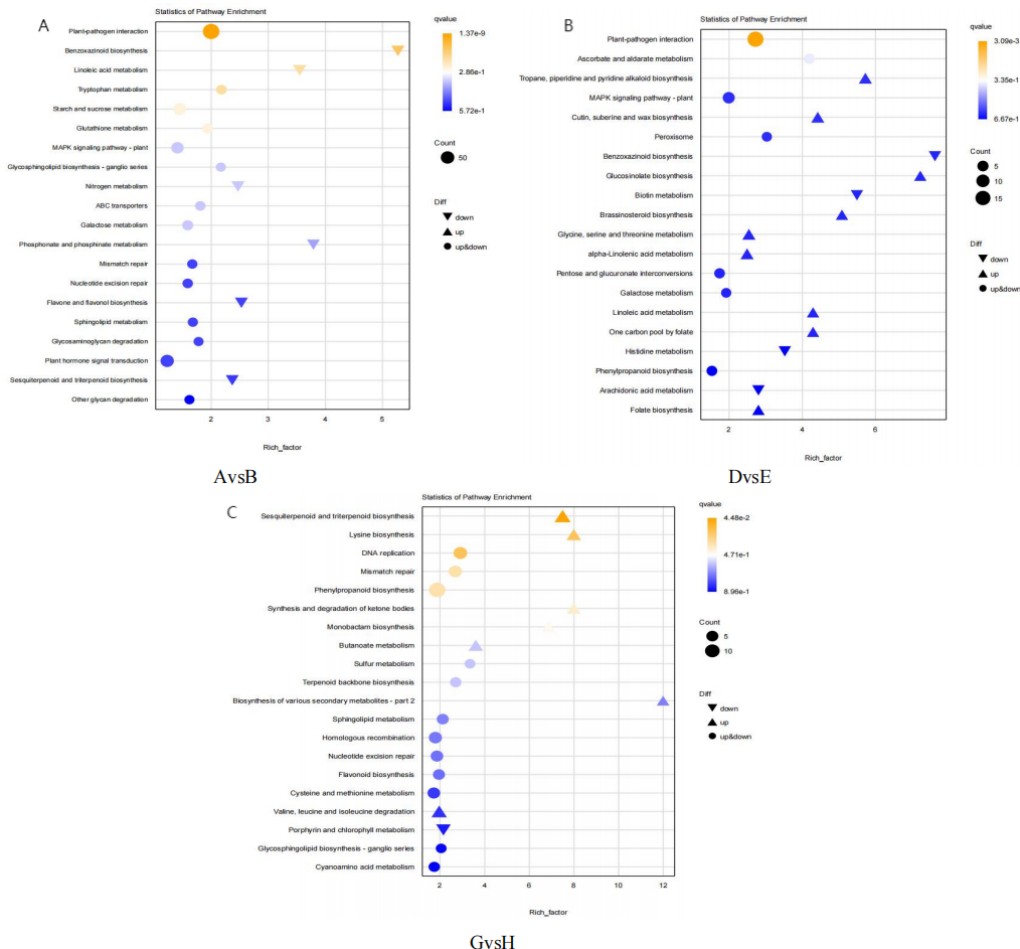

**Figure 6** **Bubble map of KEGG pathway enrichment.** (A) AvsB (P1G1 *vs* P1G3); (B) DvsE (P2G1 *vs* P2G3); (C) GvsH (P3G1 *vs* P3G3). P1: phosphorus-free treatment; P2: low phosphorus stress treatment; P3: high phosphorus supply treatment; G1: 0 µM GR24; G2: 1 µM GR24; G3: 10 µM GR24; G4: 100 µM GR24; G5: 1,000 µM GR24. The horizontal axis represents GeneRatio, which is the proportion of genes of interest annotated in the entry to the total number of differentially expressed genes. The vertical axis represents each GO annotation entry. The size of the dots represents the number of differentially expressed genes annotated in the pathway, and the color of the dots represents the *q*-value of the hypergeometric test.

GvsH, the most up-regulated transcription factor families were C2H2, bHLH, and HSF, each by one gene, while the most down-regulated families included AP2/ERF-AP2, MYB, and AP2/ERF-ERF, with decrements of four, three, and three genes, respectively.

## DISCUSSION

Phosphorus is crucial for various physiological pathways and is an essential element for plant growth (*Santoro et al., 2024*). Deficiency in phosphorus can significantly impact the morphological, physiological, and biochemical characteristics of plants. SLs have been shown to enhance plant growth indicators, contribute to the stress response (*Marzec,*

**Table 2  Major differentially expressed transcription factors.**

| TF family | AvsB | | DvsE | | GvsQ | |
|---|---|---|---|---|---|---|
| | Up | Down | Up | Down | Up | Down |
| AP2/ERF-ERF | 6 | 0 | 1 | 0 | 0 | 3 |
| C2H2 | 2 | 10 | 0 | 0 | 1 | 1 |
| MYB | 2 | 0 | 1 | 0 | 0 | 3 |
| MYB-related | 2 | 0 | 1 | 0 | 0 | 0 |
| HB-KNOX | 1 | 0 | 0 | 0 | 0 | 0 |
| NF-YB | 1 | 0 | 0 | 0 | 0 | 0 |
| PLAT2 | 1 | 0 | 0 | 0 | 0 | 0 |
| bHLH | 1 | 0 | 1 | 0 | 1 | 0 |
| zf-HD | 1 | 0 | 0 | 0 | 0 | 0 |
| NAC | 0 | 9 | 1 | 0 | 0 | 2 |
| WRKY | 0 | 8 | 3 | 0 | 0 | 0 |
| GRAS | 0 | 5 | 1 | 0 | 0 | 0 |
| bZIP | 0 | 4 | 1 | 0 | 0 | 0 |
| C3H | 0 | 2 | 0 | 0 | 0 | 2 |
| LOB | 0 | 2 | 1 | 0 | 0 | 2 |
| HB-HD-ZIP | 0 | 2 | 1 | 0 | 0 | 2 |
| HB-WOX | 0 | 0 | 0 | 0 | 1 | 0 |
| HSF | 0 | 0 | 0 | 0 | 1 | 0 |
| NBF1 | 0 | 0 | 0 | 0 | 1 | 0 |
| B3 | 0 | 0 | 1 | 0 | 0 | 0 |
| AP2/ERF-AP2 | 0 | 0 | 0 | 0 | 0 | 4 |
| B3-ARF | 0 | 0 | 0 | 0 | 0 | 2 |
| CSD | 0 | 0 | 0 | 0 | 0 | 1 |
| IWS1 | 0 | 0 | 0 | 0 | 0 | 1 |
| NEK5 | 0 | 0 | 0 | 0 | 0 | 1 |

**Notes.**

Description: (A) AvsB (P1G1 *vs* P1G3); (B) DvsE (P2G1 *vs* P2G3); (C) GvsH (P3G1 *vs* P3G3). P1: phosphorus-free treatment; P2: low phosphorus stress treatment; P3: high phosphorus supply treatment; G1: 0 μM GR24; G2: 1 μM GR24; G3: 10 μM GR24; G4: 100 μM GR24; G5: 1,000 μM GR24. The title on the left is the name of the gene family.

*2016*), and support positive growth under adverse conditions (*Andreo-Jimenez et al., 2015*; *Hong et al., 2020*; *Van Ha et al., 2014*). Additionally, GR24, a synthetic analog of SLs, has been found to specifically regulate the medicinal components of plants (*Cao et al., 2023*; *Wani et al., 2022*). Despite extensive research into GR24's ability to mitigate stress in plants, its effects on phosphorus stress in *Glycyrrhiza uralensis* Fisch have not yet been reported.

## Effect of GR24 on growth indices of *Glycyrrhiza uralensis* Fisch at different phosphorus concentrations

Phosphorus in soil is predominantly in a form that plants cannot directly absorb, leading to its immobilization and limited availability to roots, which can only access what is termed 'effective phosphorus' (*Péret et al., 2011*; *Li et al., 2023*). Consequently, root system indices such as root length and absorption area serve as indicators of plant phosphorus uptake efficiency (*Ding & Yu, 2008*). This study revealed that phosphorus stress significantly

impedes the growth morphology of *Glycyrrhiza uralensis* Fisch. These declines in growth indices were more pronounced as phosphorus stress intensified. Under conditions of low phosphorus availability, the application of GR24 at the G3 concentration significantly enhanced the fresh weight, dry weight, root length, and root projection area of *Glycyrrhiza uralensis* Fisch compared to the G1 treatment group. Supporting this observation, *Tang (2019)* reported that GR24 treatment improved the morphological indices of *Oryza sativa* L. seedlings under phosphorus stress. Additionally, *Tai et al. (2017)* found that GR24 treatment promoted biomass accumulation in *Panicum virgatum* L. seedlings under cadmium stress, corroborating our findings that Strigolactones can mitigate the detrimental effects of low phosphorus stress in plants. These results indicate that Strigolactones can enhance the growth of *Glycyrrhiza uralensis* Fisch under low phosphorus stress and improve the seedlings' adaptability to phosphorus-deficient environments. Similarly, *Pang (2020)* discovered that GR24 application promoted increases in body length and the number of lateral roots in *Astragalus membranaceus var. mongholicus* (Bunge) P.K.Hsiao. In this study, we also observed that under a high phosphorus supply, the G3 concentration of GR24 significantly enhanced the fresh weight, dry weight, root length, basal stem diameter, and root projected area of *Glycyrrhiza uralensis* Fisch. This suggests that GR24 can function as a plant hormone to regulate growth under non-stress conditions as well (*Zhou, 2016*). Under conditions of complete phosphorus deprivation, however, application of GR24 did not significantly improve the aforementioned growth indices of licorice, indicating that GR24 can only partially alleviate the adverse impacts of stress.

## Effect of GR24 on chlorophyll content of *Glycyrrhiza uralensis* Fisch at different phosphorus concentrations

Chlorophyll is essential for photosynthesis, absorbing and transferring light energy for primary photochemical reactions and other processes (*Fromme et al., 2003*; *Trevor, Vanessa & Kaiser, 2009*). The chlorophyll content in plant leaves reflects the photosynthetic capacity of the plants. Low phosphorus stress inhibits photosystem activity (*Tang, Yang & Kang, 2005*) and chloroplast membrane development, thereby reducing photosynthesis (*Li et al., 2018*). This study found that chlorophyll a and b in *Glycyrrhiza uralensis* Fisch diminished with increasing phosphorus stress, indicating a decline in photosynthetic pigments and photosynthetic performance under low phosphorus conditions. However, the exogenous application of GR24 at the G3 concentration significantly increased the content of chlorophyll a and b under low phosphorus stress. Previous studies have shown that GR24 can enhance the chlorophyll content in *Triticum aestivum* L. under drought stress (*Fang et al., 2021*), which aligns with the findings of this study. This suggests that Strigolactones may play a role in the photosynthetic regulation of plants (*Li, Tian & Shen, 2017*) and that the addition of GR24 can improve chlorophyll content in plant leaves under phosphorus stress, thereby mitigating the adverse effects of low phosphorus on photosynthetic performance (*Nagasaka et al., 2014*). Additionally, this study observed that a high supply of phosphorus inherently increases chlorophyll content. The application of an appropriate concentration of GR24 (specifically, the G3 concentration) under high phosphorus conditions also elevated the chlorophyll a and b content in *Glycyrrhiza uralensis*

Fisch, enhancing the photosynthetic performance of the plant. Contrarily, *Tian (2018)* reported that various concentrations of strigolactone had differing effects on the leaves of *Bambusa oldhamii Munro*, noting that a high concentration (5 µmol/L GR24) inhibited the green content of leaves. This discrepancy may be due to the optimal concentration of GR24 varying among different plant species.

## Effect of GR24 on antioxidant enzyme activities of *Glycyrrhiza uralensis* Fisch at different phosphorus concentrations

Phosphorus, a critical component of cell membranes, significantly influences their structure, particularly under conditions of low phosphorus stress (*Jiang et al., 2024*). To mitigate oxidative damage induced by reactive oxygen species, plants regulate the activities of SOD, POD, and CAT. This regulation is essential for protecting proteins, nucleic acids, and membrane systems, thereby supporting high growth (*Wang et al., 2022*; *Foyer & Noctor, 2013*; *Wei et al., 2018a*; *Wei et al., 2018b*). Research by *Tang (2019)* demonstrated that the application of GR24 under low phosphorus stress not only reduces the accumulation of reactive oxygen species in *Oryza sativa L.* seedlings but also enhances the activities of protective enzymes such as SOD, POD, and CAT, thereby alleviating the impacts of phosphorus stress. Further studies by *Ma et al. (2020)* and *Li et al. (2023)* found that exogenous GR24 significantly mitigates oxidative stress damage in cotton and *Malus pumila Mill.* seedlings under conditions of low-temperature and alkali stress, respectively, by increasing the activities of these antioxidant enzymes. This study also revealed that as phosphorus stress intensifies, the activities of SOD, POD, and CAT in the leaves of *Glycyrrhiza uralensis* Fisch progressively increase, indicating an inherent mechanism by which the plant alleviates the adverse effects of phosphorus deficiency. Specifically, the application of GR24 at G2, G3, and G4 concentrations enhances SOD activity, with the G3 concentration proving most effective. Similarly, GR24 at the G3 concentration notably boosts the activities of both POD and CAT. Furthermore, under high phosphorus conditions, the application of GR24 still enhances the activity of antioxidant enzymes in plants. At the G3 concentration, there is a notable increase in the activities of SOD and POD, while treatments at G2, G3, and G4 concentrations elevate CAT activity. These findings suggest that GR24 application at the G3 concentration optimally mitigates the adverse effects of phosphorus stress and enhances overall plant resilience by augmenting antioxidant enzyme activities.

## Effect of GR24 on the content of medicinal components of *Glycyrrhiza uralensis* Fisch at different phosphorus concentrations

Triterpenoids and flavonoids constitute the primary active components of *Glycyrrhiza uralensis* Fisch, with glycyrrhizic acid and glycyrrhetic acid being structurally similar triterpenoids, and liquirtigenin, liquirtin, isoliquirtigenin, isoliquirtin, and glabridin categorized as flavonoids (*Liu et al., 2013*; *Sheng et al., 2022*). Phosphorus plays a pivotal role as it is involved in the initial substrates of the terpenoid synthesis pathway, acetyl-CoA, and glyceraldehyde-3-phosphate (*Zeng et al., 2013*). Consequently, phosphorus stress impacts the production and accumulation of active ingredients in medicinal plants. Our experimental results indicate that the contents of isoliquirtigenin, liquirtigenin,

glycyrrhizic acid, isoliquirtin, and liquirtin were elevated under low phosphorus conditions compared to those in the no phosphorus treatment group. Moreover, the content of isoliquirtin was higher in both the high and low phosphorus stress conditions than in the no phosphorus stress group. This aligns with findings by *Hu et al. (2018)*, who reported that the synthesis of dihydroflavone and flavonols such as naringenin, rutin, and taxifolin could be notably inhibited under phosphorus-deficient conditions. Notably, the content of isoliquirtin decreased as phosphorus stress intensified, diverging from the results of *Winkel-Shirley (2022)*, which suggests multifaceted influences on flavonoid production in plants. Additionally, our study found that the application of GR24 under various phosphorus conditions could enhance the content of several terpenoids and flavonoids. Specifically, under no-phosphorus stress, GR24 increased the levels of glycyrrhetic acid, glycyrrhizic acid, glabridin, isoliquirtin, and liquirtin. Under high phosphorus supply, the increases were noted in isoliquirtigenin, liquirtigenin, and glycyrrhizic acid, while under low-phosphorus stress, GR24 elevated the contents of isoliquirtigenin, liquirtigenin, glycyrrhizic acid, isoliquirtin, and liquirtin. This indicates that GR24 application can variably promote the accumulation of terpenoid and flavonoid contents in *Glycyrrhiza uralensis* Fisch at different phosphorus concentrations. Such effects are consistent with previous research indicating that GR24 enhances the accumulation of anthocyanins in *Arabidopsis thaliana (L.) Heynh* (*Cao et al., 2023*) and suggests that GR24 may also modulate the production of terpenoids, as evidenced by its inhibition of diterpene secondary metabolites such as regolith methylesterase and strigolactone in *Tripterygium wilfordii Hook. f.* suspension cells (*Wu et al., 2019*).

## Effect of GR24 on the transcriptome of *Glycyrrhiza uralensis* Fisch at different phosphorus concentrations

Plants exhibit a sophisticated response to adversity that encompasses physiological, biochemical, and metabolic processes. This response is mediated through the synergistic actions of multiple genes and a complex mechanism of co-regulation (*Li et al., 2024*; *Pant et al., 2015*; *Sun et al., 2016*). In this study, we identified 1,298, 163, and 513 DEGs under conditions of no-phosphorus stress, low-phosphorus stress, and high phosphorus supply, respectively. Notably, the lowest number of DEGs was observed under low-phosphorus stress, aligning with findings by *Guan et al. (2023)*. GO enrichment analysis revealed that the DEGs across the three treatments were predominantly associated with cellular processes, metabolic processes, cellular anatomical entities, and catalytic activity. KEGG pathway analysis showed that the no-phosphorus and low-phosphorus stress treatments were significantly enriched in plant-pathogen interaction and signaling pathways, respectively. Conversely, the high phosphorus supply treatment was significantly enriched in pathways such as phenylpropanoid biosynthesis and DNA replication. The phenylpropanoid biosynthesis pathway, crucial for synthesizing secondary metabolites, also plays a vital role in plant growth, development, and environmental adaptation (*Zhong & Ye, 2009*), thereby underscoring the regulatory effects of GR24 on *Glycyrrhiza uralensis* Fisch. Moreover, all three treatments showed enrichment in pathways related to secondary

metabolite production, indicating a positive effect of GR24 on the synthesis of medicinal components in Ural *Glycyrrhiza uralensis* Fisch.

Our findings demonstrate that transcription factors such as bHLH, AP2, MYB, WRKY, and NAC are responsive to phosphorus deprivation. Specifically, bHLH (Wang et al., 2023) and AP2 (*Zhao et al., 2018*) transcription factors not only participate in plant growth and development but also respond to secondary metabolic processes and abiotic stresses (*Zhang, 2023*). Under no phosphorus stress, the most differentially expressed transcription factors were from the C2H2 family, followed by the NAC family, all of which were down-regulated. This suggests that GR24 application may enhance tolerance to phosphorus deficiency in *Glycyrrhiza uralensis* Fisch. Under low phosphorus stress, the WRKY family was the most differentially expressed and predominantly up-regulated, consistent with observations by *Huang et al. (2012)* that the WRKY family exhibits varied expression patterns in response to different abiotic stresses. Under high phosphorus supply, the AP2/ERF-ERF family was the most differentially expressed transcription factor group and was down-regulated. This down-regulation might be attributed to the effect of GR24 on growth hormone regulation in *Glycyrrhiza uralensis* Fisch, impacting associated signaling pathways (*Feng et al., 2020*; *Gu et al., 2017*; *Ritonga et al., 2021*).

## CONCLUSIONS

Overall, the application of GR24 ameliorates the adverse effects of phosphorus stress in *Glycyrrhiza uralensis* Fisch by increasing biomass, photosynthetic pigment levels, and while reducing the content of antioxidant enzymes and enhancing the accumulation of certain triterpenoids and flavonoids. Differential expression genes (DEGs) were primarily enriched in pathways favorable to the growth, development, and secondary metabolite accumulation in *Glycyrrhiza uralensis* Fisch, with the upregulation of the WRKY family related to phosphorus stress response. The G3 concentration was found to be most effective, indicating that GR24 application improves the phosphorus deficiency tolerance of *Glycyrrhiza uralensis* Fisch. Moreover, under high phosphorus conditions, GR24 promotes plant growth and development, increases biomass and, elevates chlorophyll content, decreases antioxidant enzyme levels, and enhances the accumulation of medicinal components. DEGs in this context are mainly enriched in pathways related to plant growth regulation, and elevate the AP2 family of transcription factors associated with the plant growth hormone signaling pathway, demonstrating that GR24 not only improves growth under low phosphorus conditions but also positively impacts the growth and development of *Glycyrrhiza uralensis* Fisch under high phosphorus nutrition.

## ACKNOWLEDGEMENTS

This research forms part of the author's master's thesis.

### Funding

This study was supported by the National Natural Science Foundation of China (31871568,31560656), the Project for Innovative Reasearch Team of Chengdu Normal University (CSCXTD2020A04), and the Innovation and Entrepreneurship Project for college students in Sichuan Province (S202214389093). The funders had no role in study design, data collection and analysis, decision to publish, or preparation of the manuscript.

### Grant Disclosures

The following grant information was disclosed by the authors:
National Natural Science Foundation of China: 31871568, 31560656.
Project for Innovative Reasearch Team of Chengdu Normal University: CSCXTD2020A04.
Innovation and Entrepreneurship Project for college students in Sichuan Province: S202214389093.

### Competing Interests

The authors declare there are no competing interests.

### Author Contributions

- Yuting Jing conceived and designed the experiments, performed the experiments, analyzed the data, prepared figures and/or tables, and approved the final draft.
- Man Li analyzed the data, prepared figures and/or tables, and approved the final draft.
- Yong Wu analyzed the data, authored or reviewed drafts of the article, and approved the final draft.
- Chengming Zhang analyzed the data, authored or reviewed drafts of the article, and approved the final draft.
- Chengshu Qiu analyzed the data, authored or reviewed drafts of the article, and approved the final draft.
- Hengming Zhao performed the experiments, authored or reviewed drafts of the article, and approved the final draft.
- Li Zhuang conceived and designed the experiments, authored or reviewed drafts of the article, and approved the final draft.
- Hongling Liu conceived and designed the experiments, authored or reviewed drafts of the article, and approved the final draft.

### DNA Deposition

The following information was supplied regarding the deposition of DNA sequences:
The *Glycyrrhiza uralensis* raw sequence reads are available at NCBI: PRJNA1112769.

### Data Availability

The phosphorus standard curves and HPLC parameters are available in the Supplementary File.

## Supplemental Information

Supplemental information for this article can be found online at http://dx.doi.org/10.7717/peerj.18546#supplemental-information.

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
