# Peer review of "Effect of GR24 on the growth and development of licorice under low phosphorus stress"

_PeerJ, doi:10.7717/peerj.18546_

## Round 0.1 · original submission · Major Revisions

This manuscript reads like a rough draft. The manuscript requires major revisions to address the numerous grammatical errors, improve clarity and organization, enhance methodological details, provide immediate interpretation of results, deepen the discussion, better reference figures and tables, address limitations in the conclusion, and ensure consistent formatting.

Some examples include, the lack of figure legends, notation like "uM/L" (which makes no sense), lack of adequate description of methods, lack of explanation of replicates/statistical methods, incomplete supplementary data (no data for transcriptional profiling; the other data has no explanation in the spreadsheets), scientific names are not italicized, some of the references are incomplete, etc. My suggestion is that the authors rewrite the manuscript without substantial improvements, the manuscript falls short of the standards required for publication. If resubmitted, it will need to be significantly improved.

Reviewer 1 ·

Basic reporting

Glycyrrhiza is a perennial herbaceous medicinal plant with extensive applications in the pharmaceutical industry. The growth of Glycyrrhiza is often limited by soil phosphorus availability, as most of China's arable land is in a phosphorus deficient state. This study used Ural Glycyrrhiza uralensis Fisch as the research object and applied GR24 in three simulated environments: simulating no phosphorus (P1), low phosphorus (P2), and normal phosphorus supply (P3). Research the effects of GR24 on the growth and development of licorice through physiological and biochemical indicators, active ingredient content, transcriptomics and other methods. And obtain the optimal application concentration of GR24, providing a theoretical basis for licorice cultivation. However, this manuscript still has certain shortcomings, such as large data errors in some experiments, the figures are not clear and comprehensive, and inaccurate expression and grammar errors throughout the manuscript. This manuscript has not yet reached the level of publication, and the specific suggestions are as follows.
Title: The current title is different phosphorus concentrations, but this study has few treatment groups, and the entire manuscript mentions low phosphorus stress. It is recommended to change the title to “Effect of GR24 on the growth and development of licorice under low phosphorus stress”.
Introduction: Line 76 mentions Solanum lactone, and line 127 mentions synthetic monocrotaline oligolactone. The relationship between these two compounds and GR24 is not clearly explained.
Method: Line 123 NHH42PO4? Lines 127-128 "the treatment group sprayed with GR24 (synthetic monocrotaline strigolactone) as a blank control at four concentration levels...". Which is the control group? Line130 "until the foliage surface congealed into The frequency was once a week", congealed into what?
Result: Lines 290, 292-293 "Biological process (BP), Cellular component (CC)...", the abbreviations are repeated. Line 314 Plant Pathogen Interaction (Plant Pathogen Interaction) is repeated twice.
Fig. 1: The P2G5 treatment group in P accumulation in stem showed a significant increase in results compared to other groups. The author should exclude individual differences caused by sampling and suggest explaining them in the discussion.
Fig. 4: The glycyrrhetic acid content data error in P1G3 treatment is too large. It is recommended to add a few duplicate samples to this treatment.
Fig. 7: The cellular component and molecular function have no vertical axis.
Fig. 8: The figure is unclear and the words (gene_number) in the figure are incomplete.
Discussion: What is the abbreviation of line 338 "SLs"? Please indicate clearly. Line 357-358 "Phosphorus stress can allocate the damage caused by licorice at SLs", the expression is not accurate, it should be "SLs can allocate the damage of low phosphorus stress to licorice".
The English language should be improved to ensure that an international audience can clearly understand your text.

Experimental design

According to this title, more treatment groups with different concentrations of phosphorus should be set up.

Validity of the findings

There are large errors in the data, poor repeatability between samples, and some data lack ordinate.

Reviewer 2 ·

Basic reporting

The manuscript titled " Effect of GR24 on the growth and development of licorice under different phosphorus concentrations" requires several revisions to enhance its clarity and scientific rigor:
1. It is essential to italicize all scientific names throughout the manuscript to confirm to standard scientific writing conventions.
2. The abstract should be revised to clarify the meanings of all abbreviations used, ensuring that the content is accessible to a broad audience.
3. Abstract should be divided into subheadings as rules of journal.
4. In line 83, when referring to "the following table," please specify which specific table contains the summarized results to avoid ambiguity.
5. The manuscript lacks references detailing the experimental design, which is crucial for understanding the methodology employed.
6. Numerous grammatical errors are present throughout the text, requiring careful proofreading and correction.
7. All protocols used in the study must be referenced to provide transparency and allow for reproducibility of the findings.
8. It is important to clearly indicate in Table 1 whether means ± SD or means ± SE are reported to facilitate accurate interpretation by readers.
9. The conclusion section is currently brief and would benefit from a more comprehensive summary of the study's findings and implications.
10. References are missing in the initial part of the discussion, where citations are needed to support and contextualize the findings.
11. The formula used for measuring chlorophyll content should be explicitly stated, as no formula was written in the manuscript.

Experimental design

Although experiments were performed well but no references were given at the end.

Validity of the findings

Data is novel and original.
Conclusion section is too short.

---

## Round 0.2 · Minor Revisions

The manuscript is improved but still needs work to address some outstanding issues. I have attached a copy of the tracked manuscript with some of my comments. Avoid referring to the P3 phosphate concentration as normal, better to refer to it as high relative to the other concentrations. Please address these minor issues. Also, please provide the units for the chlorophyll measurements in the excel supplementary data, define the units for the SOD etc measurements. You also need to have complete figure legends. Figures and tables should be stand alone and a reader should be able to understand the figure without referring to the manuscript text.

---

## Round 0.3 · Minor Revisions

Please include the figure legends. I still can't find appropriate legends for each of the figures.

---

## Round 0.4 · Minor Revisions

You have figure titles not legends. A figure legend describes and explains all aspects of a figure. A reader should be able to read a figure legend and understand what it is showing without having to refer back to the main text. They should be stand alone.

---

## Round 0.5 · Minor Revisions

Please adjust the figure legends so that they are descriptive. Consult figure legends from any publication to understand how figures are typically explained. For example, figure legends do not have the term description:…. This seems like it might be AI generated. Please assign a letter to different panels in the same figure - as in Figure 1A, Figure 1B etc.

---

## Round 0.6 · accepted · Accept

Thank you for making the changes. The word “description” should be removed in the final version of the figure legends.